# Understanding the Operating Mode of Fe$^0$/Fe-Sulfide/H$_2$O Systems for Water Treatment

**Minhui Xiao [1], Rui Hu [1,*], Xuesong Cui [1], Willis Gwenzi [2] and Chicgoua Noubactep [1,3,4,*]**

[1] School of Earth Science and Engineering, Hohai University, Fo Cheng Xi Road 8, Nanjing 211100, China; xiaominhui@hhu.edu.cn (M.X.); cuixuesong@hhu.edu.cn (X.C.)

[2] Biosystems and Environmental Engineering Research Group, Department of Soil Science and Agricultural Engineering, Faculty of Agriculture, University of Zimbabwe, P.O. Box MP167, Mount Pleasant, Harare, Zimbabwe; wgwenzi@yahoo.co.uk

[3] Department of Applied Geology, University of Göttingen, Goldschmidtstraße 3, D-37077 Göttingen, Germany

[4] Department of Water and Environmental Science and Engineering, Nelson Mandela African Institution of Science and Technology, P.O. Box 447, Arusha, Tanzania

\* Correspondence: rhu@hhu.edu.cn (R.H.); cnoubac@gwdg.de (C.N.); Tel.: +49-551-393-3191 (C.N.)

**Abstract:** The general suitability of water treatment systems involving metallic iron (Fe$^0$) is well-established. Various attempts have been made to improve the efficiency of conventional Fe$^0$ systems. One promising approach combines granular Fe$^0$ and an iron sulfide mineral to form Fe$^0$/Fe-sulfide/H$_2$O systems. An improved understanding of the fundamental principles by which such systems operate is still needed. Through a systematic analysis of possible reactions and the probability of their occurrence, this study establishes that sulfide minerals primarily sustain iron corrosion by lowering the pH of the system. Thus, chemical reduction mediated by Fe$^{II}$ species (indirect reduction) is a plausible explanation for the documented reductive transformations. Such a mechanism is consistent with the nature and distribution of reported reaction products. While considering the mass balance of iron, it appears that lowering the pH value increases Fe$^0$ dissolution, and thus subsequent precipitation of hydroxides. This precipitation reaction is coupled with the occlusion of contaminants (co-precipitation or irreversible adsorption). The extent to which individual sulfides impact the efficiency of the tested systems depends on their intrinsic reactivities and the operational conditions (e.g., sulfide dosage, particle size, experimental duration). Future research directions, including the extension of Fe$^0$/Fe-sulfide/H$_2$O systems to drinking water filters and (domestic) wastewater treatment using the multi-soil-layering method are highlighted.

**Keywords:** contaminant removal; iron corrosion; pyrite oxidation; synergetic effects; zero-valent iron

## 1. Introduction

Metallic iron (Fe$^0$) is a readily available and low-cost reactive material, industrially used in water treatment since the second half of the 19th Century [1–6]. Fe$^0$ has been successfully used to remove color and various classes of biological and chemical pollutants from the aqueous phase [1,3,4,7–9]. During the 1990s, Fe$^0$ has been rediscovered and successfully applied for environmental remediation, including in subsurface permeable reactive barriers (Fe$^0$ PRBs) [10–18]. New Fe$^0$-based technologies for safe drinking water supply were then derived from the Fe$^0$-based PRB technology [19–28].

The efficiency of a Fe$^0$/H$_2$O remediation system for water treatment has been observed to be impaired by an inherent characteristic of aqueous iron corrosion at near-neutral pH values, which is often termed "reactivity loss" [10,29]. Reactivity loss is characterized by the precipitation of iron

(hydr)oxides in the vicinity of or at the $Fe^0$ surface. The layer of (hydr)oxides or oxide scale acts as a physical and electronic barrier that "passivates" the material, thereby significantly reducing the efficiency of the $Fe^0$-based system [10,18,30–32]. In essence, "reactivity loss" is a misleading term, as reactivity is an intrinsic property of each material and cannot change with operational variables or under given operating conditions [28,33–36]. That is the reason why it is restored by appropriate treatments (e.g., acid wash) [15,37] and/or sustained by other additive materials (e.g., using $FeS_2$ or $MnO_2$) [29,38–44]. Despite its misleading nature, the term "reactivity loss" is partly maintained here, for the sake of clarity. Several tools have been introduced to address "reactivity loss" [15,37,44–48], including the amendment of granular $Fe^0$ with granular iron sulfide minerals, which is the focus of the current study. Other tools as summarized by Lü et al. [18] include using the following: fabrication of (dispersed) nanoscale $Fe^0$, multi-metallic materials, weak magnetic field and the fabrication of several $Fe^0$-based composites.

Lipczynska-Kochany et al. [38] was probably the first research group to use iron sulfide minerals to sustain the $Fe^0$ reactivity, thereby increasing the efficiency of conventional $Fe^0/H_2O$ systems for water treatment. Accordingly, during the past 25 years, a large number of scientific publications have reported on the enhanced efficiency of $Fe^0/Fe$-sulfide/$H_2O$ systems relative to their $Fe^0/H_2O$ counterparts [18,29, 44,47–53]. However, the mechanisms for this enhancement are still to be established [18,44,52]. Clearly, there is still controversy on the reasons why the presence of pyrite enhances the efficiency of $Fe^0/H_2O$ systems for contaminant removal. This troublesome situation is not a good basis for a science-based design of the next generation $Fe^0/Fe$-sulfide/$H_2O$ system, and the development of the $Fe^0$ technology in general. Table 1 suggests that the main cause for this situation lies in the various different experimental designs and conditions for the reported investigations [53–57]. It is seen that the used $Fe^0$ and $FeS_2$ dosages (in g $L^{-1}$) vary largely, while the mixing intensities vary from 0 to 400 rpm.

**Table 1.** Comparison of literature data on experimental conditions for investigating the $Fe^0/FeS_2/H_2O$ system. Studies were performed at initial pH values ranging from 3.0 to 10.0, and $Fe^0$ mass loadings ranging from 0.2 to 200 g $L^{-1}$. The size of pyrite and $Fe^0$ also showed large variations among studies. "n.s." stands for not specified, the authors have referred to Supporting Information.

| Contaminant Nature | Pyrite | | | | Metallic Iron | | Stirring | Ref. |
|---|---|---|---|---|---|---|---|---|
| | $pH_0$ (-) | V (mL) | d (mm) | $\rho$(g $L^{-1}$) | d (mm) | $\rho$(g $L^{-1}$) | (rpm) | |
| $CCl_4$ | 6.0 | 25 | <0.841 | 200 | 0.150 | 200 | 170 | 38 |
| Arsenic | 3.0 to 9.0 | 500 | n.s. | 0.20 or 2.0 | n.s. | 0.20 or 2.0 | 400 | 53 |
| Uranium | 7.2 | 20 | 200 to 630 | 25 | 1.6 to 2.5 | 15 | 0.0 | 54, 55 |
| Nitrobenzene | 5.0 to 10.0 | 150 | 40 to 75 | 0.5 to 3.0 | 40 to 75 | 0.5 | 200 | 52 |
| Orange II | 7.0 | 150.0 | 38 to 50 | 0.25 or 2.0 | 0.25 to 2.0 | 0.25 or 0.50 | 200 | 47 |
| RR X-3B | 7.0 | 150.0 | 38 to 50 | 0.25 or 2.0 | 0.25 to 2.0 | 0.25 or 0.50 | 200 | 47 |
| Amido Black 10B | 7.0 | 150.0 | 38 to 50 | 0.25 or 2.0 | 0.25 to 2.0 | 0.25 or 0.50 | 200 | 47 |
| Methylorange | 6.9 | 22 | 38 to 50 | 0.25 or 2.0 | 1.0 | 0.5 | 0.0 | 57 |
| Methylene blue | 7.0 | 22 | 38 to 50 | 0.25 or 2.0 | 1.0 | 0.5 | 0.0 | 57 |

The objectives of the current study are: (i) to critique and clarify the role of sulfide minerals in enhancing the efficiency of $Fe^0/H_2O$ systems on a purely analytical basis, and (ii) to highlight the key knowledge gaps and future research directions. The paper is structured as follows; first, the chemistry of the $Fe^0/Fe$-sulfide/$H_2O$ system will be presented. Then the mechanisms of contaminant removal are discussed. Finally, the key knowledge gaps and future directions are then highlighted.

## 2. The $Fe^0/Fe$-Sulfide/$H_2O$ System

When a reactive Fe-sulfide (e.g., $FeS_2$) is immersed in aqueous systems under oxic conditions (presence of $O_2$), it is oxidatively dissolved and each mole of $FeS_2$ produces four moles of protons ($H^+$)

(Equation (1)). Under anoxic conditions (absence of $O_2$), pyrite dissolution produces four times more protons (Equation (2)).

$$4\,FeS_2 + 15\,O_2 + 10\,H_2O \Rightarrow 4\,FeOOH + 16\,H^+ + 8\,SO_4{}^{2-} \tag{1}$$

$$FeS_2 + 14\,Fe^{3+} + 8\,H_2O \Rightarrow 15\,Fe^{2+} + 2\,SO_4{}^{2-} + 16\,H^+ \tag{2}$$

Reactions (1) and (2) show that the oxidative dissolution of pyrite always generates protons, and hence induces an acidification of non-buffered aqueous systems (Figure 1). In this study, Fe-sulfides mixed with $Fe^0$ are considered, wherein aqueous oxidative dissolution is a proton-consuming process (Equation (3)). Reaction 3 implies that $Fe^0$ is oxidized by protons generated by the hydrolysis of water ($H_2O \Leftrightarrow H^+ + OH^-$). The abundance of water (solvent) implies that reaction 3 should never be assumed as a side reaction, unless it is demonstrated otherwise. However, for natural waters, relevant pollutants are typically present in trace amounts and the pH value is close to 7 [54].

$$Fe^0 + 2\,H^+ \Rightarrow Fe^{2+} + H_2 \tag{3}$$

Depending on $O_2$ availability, reactions (1) and (3) or reactions (2) and (3) compete for controlling the pH of the system. Thus, the final pH value depends on at least four factors: (i) the relative dosages of $Fe^0$ and $FeS_2$, (ii) their respective intrinsic reactivities, (iii) the contact time and (iv) the temperature. In other words, the trivial pH decrease with increasing $FeS_2$ loadings (Figure 1) cannot be directly correlated with the extent of contaminant removal.

One hypothetical case is the one in which $FeS_2$ oxidation initially dominates [55]. In such as case, the pH value first decreases, and then progressively increases until it reaches a final value. Acidity is first produced by reaction 1 or/and reaction 2, then progressively consumed by reaction 3 until a pseudo-steady state is reached. Considering that $Fe^0$ and $FeS_2$ have long-term reactivity, investigating $Fe^0$/Fe-sulfide/$H_2O$ systems in batch mode under quiescent conditions can last for several weeks before reaching a pseudo-equilibrium [56]. During this long time of reactivity, weathering of in-situ generated iron (hydr)oxides (corrosion products) through acidification (Equation (1) or Equation (2)) also occurs, but will not be discussed further herein. Note that reactions 1 and 2 are not mutually exclusive of each other—rather, they may occur simultaneously to some extent. An initially closed oxic system will turn anoxic over time because both pyrite oxidation (Equation (1)) and iron corrosion (Equation (3)) are $O_2$ scavengers. The $O_2$ scavenging nature of iron corrosion is due to the fact that $Fe^{2+}$ (Equation (3)) is readily oxidized by dissolved $O_2$ (Equation (4)). Clearly, iron corrosion (Equation (5)) is accelerated because $Fe^{2+}$ is consumed (Le Chatelier's principle). In other words, $Fe^0$ is not oxidized by $O_2$ (Equation (6)) or any contaminant, as commonly reported in the $Fe^0$ literature (Equation (7)), but is oxidized in the presence of oxygen (Equations (3) and (4)).

$$4\,Fe^{2+} + O_2 + 2\,H^+ \Rightarrow 4\,Fe^{3+} + 2\,HO^- \tag{4}$$

$$Fe^0 \Leftrightarrow Fe^{2+} + 2\,e^- \tag{5}$$

$$2\,Fe^0 + O_2 + 2\,H_2O \Rightarrow 2\,Fe^{2+} + 4\,OH^- \tag{6}$$

$$Fe^0 + RX + H^+ \Rightarrow Fe^{2+} + RH + X^- \tag{7}$$

where X represents a halogen (e.g., Cl).

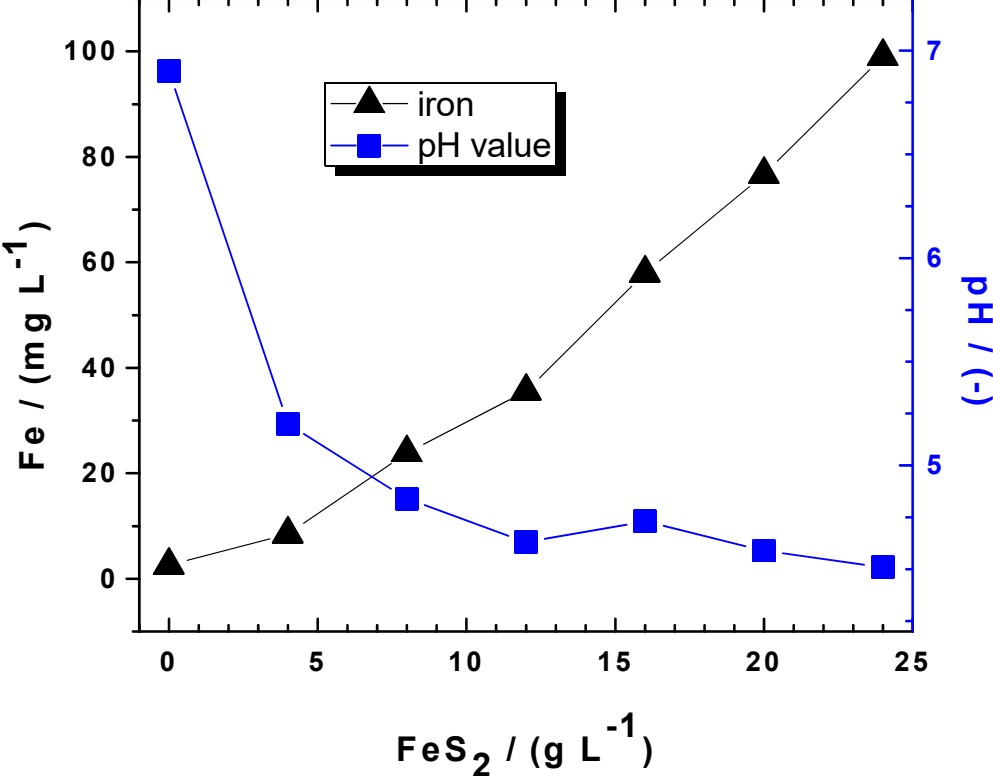

**Figure 1.** Illustration of the pH shifting property of pyrite ($FeS_2$) for the $Fe^0/H_2O$ system. The initial pH value is 7.0; data are from Cui [57]. The lower the $FeS_2$ loading, the higher the pH value. Correlating the final pH values to the extent of decontamination depends on other operational parameters, including the experimental duration. The represented lines are not fitting functions; they just join the points to facilitate visualization.

The presentation until here reveals the complexity of a contaminant-free $Fe^0/Fe$-sulfide/$H_2O$ system. It also shows the possible variabilities of systems commonly termed "Fe-sulfide enhanced $Fe^0/H_2O$." Regardless of the presence of any contaminant, an Fe-sulfide enhanced $Fe^0/H_2O$ system represents the at rest state of dynamic processes, implying that both $Fe^0$ and Fe-sulfide produce reducing species (e.g., $Fe^{II}$, $H_2$) and adsorbing agents (iron hydroxides and oxides). Moreover, during the precipitation of hydroxides, contaminants are enmeshed or occluded in their matrix and also co-precipitated [58,59]. In other words, in $Fe^0/Fe$-sulfide/$H_2O$ systems, contaminants are more or less removed by adsorption and co-precipitation, while some reducible contaminants are probably quantitatively reduced. However, two key issues should be considered: (i) observed reductive transformations are not the cathodic reactions coupled to iron dissolution (Equation (5)), and (ii) chemical reduction is not a relevant removal mechanism for many contaminants at the concentration ranges of natural waters [28,57,60,61].

The inherent limitation of chemical methods for the treatment of natural water arises from the fact that the residual concentration based on the solubility limit is still too high compared to an admissible maximum concentration level (MCL). For example, the residual concentrations of the water treatment chemicals (e.g., chlorine, $Al^{3+}$) may exceed the MCL for drinking water set by the World Health Organization (WHO). As an example, using $Ca^{2+}$ addition to lower the concentration of fluoride ($F^-$) from polluted water yields an equilibrium concentration of 8.0 mg $L^{-1}$ ([$F^-$]), which is too high compared to the WHO MCL value of 1.5 mg $L^{-1}$. To reach values less than 1.5 mg $L^{-1}$, physical methods are needed (e.g., adsorption, dilution or water blending, ion-exchange and size-exclusion), $Fe^0$ filters were proven a poorly-efficient alternative [62]. For organic species, the reduction products must be removed from water as well, particularly in the context of safe drinking water provision. For environmental remediation, it may suffice if the reaction products are biodegradable [10]. Clearly, one

major problem of the research on using $Fe^0$ for water treatment has been to randomly interchange the terms "contaminant reduction" and "contaminant removal" as if reduced species are automatically removed [28,33,34]. It has been constantly pointed out that no satisfying mass balance of the species involved, including $Fe^{0,}$ has been presented [63–65].

The thermodynamic arguments given in this section are sufficient to definitively rule out direct reduction (electrons from $Fe^0$) as a mechanism of contaminant transformation in $Fe^0/H_2O$ systems as a rule. Documented reductive transformations are mediated by primary reducing species ($Fe^{II}$, $H/H_2$) and secondary corrosion products (e.g., $Fe_3O_4$, green rust) operating in synergy. By inducing a timely pH shift, the addition of Fe-minerals just intensifies the discussed processes. Since the first mechanistic work of Matheson and Tratnyek [66], reactions similar to Equation (7) were being written as the first choice and are mostly considered as scientifically established. Moreover, the stoichiometry of reaction 7 is used to model/design $Fe^0/H_2O$ systems [28,32,36,67].

O'Hannesin and Gillham [68] acknowledged that the adoption of the reductive transformation concept was a "broad consensus" among researchers in the $Fe^0$ remediation community. This raises a question: how can a flawed consensus survive for two decades without being addressed? There are at least two possible answers: (i) it has not been really questioned or investigated by the majority of active researchers; and/or (ii) research findings pointing to the flaws of the concept have been overlooked or refuted. A number of cases exist to support these explanations. For example, despite a number of short communications and review articles challenging the validity of the concept (Table 2), several subsequent studies have perpetuated the flawed concept to explain the removal of contaminants by $Fe^0/H_2O$. This situation has prompted the publication of a "review of reviews" in 2015 in Water Research [64], but the large majority of researchers are still propagating the false view. The most referenced article with 242 citations is from 2008 (12 years old). During this period some thousands of scientific articles have been published on the remediation $Fe^0/H_2O$ system. The next section is focused on the $Fe^0/Fe$-sulfide/$H_2O$ system.

**Table 2.** Selected articles disproving the reductive transformation concept for the $Fe^0/H_2O$ system and their current bibliometry according to Scopus and ScienceDirect (www.scopus.com: 23/03/2020). Self-citation is included. It is seen that the best citation rate is 20 per year in a context where more than 500 articles are produced every year.

| Title | Journal | Year | Citation | Citation/Year |
|---|---|---|---|---|
| Processes of contaminant removal in "$Fe^0$–$H_2O$" systems revisited. The importance of coprecipitation. | Open Environ. Sci. | 2007 | n.a. | n.a. |
| A critical review on the mechanism of contaminant removal in $Fe^0$–$H_2O$ systems | Environ. Technol. | 2008 | 242 | 20.2 |
| $Fe^0$-based alloys for environmental remediation: Thinking outside the box | J. Hazard. Mater. | 2009 | 23 | 2.1 |
| An analysis of the evolution of reactive species in $Fe^0/H_2O$ systems | J. Hazard. Mater. | 2009 | 108 | 9.8 |
| On the operating mode of bimetallic systems for environmental remediation | J. Hazard. Mater. | 2009 | 29 | 2.6 |
| On the validity of specific rate constants (kSA) in $Fe^0/H_2O$ systems | J. Hazard. Mater. | 2009 | 14 | 1.6 |
| On nanoscale metallic iron for groundwater remediation | J. Hazard. Mater. | 2010 | 45 | 4.5 |
| The fundamental mechanism of aqueous contaminant removal by metallic iron | Water SA | 2010 | 125 | 12.5 |
| The suitability of metallic iron for environmental remediation | Environ. Prog Sustain. | 2010 | 55 | 5.5 |

**Table 2.** *Cont.*

| Title | Journal | Year | Citation | Citation/Year |
|---|---|---|---|---|
| Aqueous contaminant removal by metallic iron: Is the paradigm shifting? | Water SA | 2011 | 59 | 6.6 |
| Metallic iron for environmental remediation: Back to textbooks | Fresenius Environ. Bull. | 2012 | 13 | 1.6 |
| Flaws in the design of $Fe^0$-based filtration systems? | Chemosphere | 2014 | 24 | 4.0 |
| Metallic iron for environmental remediation: A review of reviews | Water Research | 2015 | 72 | 14.4 |
| No scientific debate in the zero-valent iron literature | CLEAN—Soil, Air, Water | 2016 | 8 | 2.0 |
| Research on metallic iron for environmental remediation: Stopping growing sloppy science | Chemosphere | 2016 | 21 | 5.3 |
| Predicting the hydraulic conductivity of metallic iron filters: Modeling gone astray | Water | 2016 | 18 | 4.5 |
| Metallic iron for water treatment: Leaving the valley of confusion | Applied Water Science | 2017 | n.a. | n.a. |
| Rescuing $Fe^0$ remediation research from its systemic flaws | Res. Rev. Insights | 2017 | n.a. | n.a. |
| Metallic iron for environmental remediation: How experts maintain a comfortable status quo | Fresenius Environ. Bull. | 2018 | n.a. | n.a. |
| Iron corrosion: Scientific heritage in jeopardy | Sustainability | 2018 | 5 | 2.5 |

## 3. Contaminant Removal in $Fe^0$/Fe-Sulfide/$H_2O$ Systems

Fe-sulfides have been successfully used as stand-alone reducing agents for many dissolved organic and inorganic species [69–79]. It is one of the most powerful natural reducing agents and has been demonstrated to sustain the environmental redox cycling of manganese [80,81]. Therefore, mixing Fe-sulfides and $Fe^0$ corresponds to mixing two reducing agents, and should result in even more reducing systems (without considering the pH shift capacity of Fe-sulfides). Yet studies have been published comparing the relative kinetics and extent of contaminant removal in the following three systems: (i) Fe-sulfide alone, (ii) $Fe^0$ alone and (iii) the $Fe^0$/Fe-sulfide mixture [18,47,48,52,53,55,56,82]. For example, Lü et al. [18] observed no nitrobenzene removal in the pure $FeS_2$ system, moderate removal in the pure $Fe^0$ system and increased removal in the $Fe^0$/$FeS_2$ system. Noubactep et al. [56] reported exactly the same trend for $U^{VI}$ removal. These results are correctly interpreted as an indication of the synergetic effect of $FeS_2$ (or more generally Fe-sulfides) on contaminant removal by $Fe^0$ [44,52,56]. However, the role of Fe-sulfides is limited to inducing pH shift, and thus increasing corrosion, at least during the lag time of the experimental durations ranging from 300 min for Lü et al. [18] to 120 days for Noubactep et al. [56]. It is of critical importance to state that well-documented reductive transformation of both contaminants in the pure $FeS_2$ system [76,83] were not observed by both research groups. One reason for this is certainly the slow kinetics of $Fe^{II}$-mediated reduction processes [66,84]. Accordingly, both research groups independently demonstrated that the presence of Fe-sulfides improve contaminant removal in $Fe^0$/$H_2O$ systems.

In $Fe^0$/$FeS_2$ systems, the primary role of $FeS_2$ is to induce acidification. The process is accelerated by the LeChatelier's principle as generated protons (Equations (1) and (2)) are used for $Fe^0$ corrosion (Equation (3)). Generated $Fe^{II}$ species are not available in the vicinity of the mineral to induce the surface-catalyzed reductive transformation observed in nature and reproduced in the laboratory (in the absence of $Fe^0$) [76,80,81,83,85]. It should be recalled that permeable reactive barriers (PRBs) of Fe-sulfides are a stand-alone tool for groundwater remediation [85], and have been even suggested as alternatives to $Fe^0$ PRBs because they are less prone to clogging [10,86].

As stated earlier, amending $Fe^0/H_2O$ systems with Fe-sulfides was a tool to increase its efficiency (often termed reactivity) for contaminant removal. The question is, "which efficiency?" or "the efficiency to treat which type of natural water?" In essence, the development of the $Fe^0$ technology during the past three decades has been a sort of a race for the development of the most reactive $Fe^0$ material. However, given that each natural water is unique in terms of physico-chemical and biological contaminants, it is unlikely that any single material (be it the most reactive one) would be the most appropriate for all situations. On the contrary, using a more reactive material at a certain site would result in early $Fe^0$ depletion or early system clogging [87]. Thus, amending a given $Fe^0/H_2O$ system with Fe-sulfides should be decided on a site-specific basis. Moreover, which Fe-sulfide or Fe/Fe-sulfide to use, and which amount (proportion or mix ratio) thereof should be a rationale decision based on the fundamental understanding of the reactivity or corrosion kinetics of the materials. Therefore, the further development of the $Fe^0$ technology calls for the characterization and subsequent establishment of a database of the intrinsic reactivities of $Fe^0$ materials and reactive additives (e.g., $FeS_2$, $MnO_2$).

## 4. Characterizing the Efficiency of $Fe^0/FeS_2$ Systems

Up to this point, the presentation shows the potential of Fe-sulfides to render $Fe^0$ systems more efficient and sustainable than conventional $Fe^0/H_2O$ systems (i.e., without reactive additives). However, available results are collectively qualitative, as they were achieved under very different operating conditions (Table 1), and in some cases, relevant data to interpret the presented results are missing. For example, most works characterizing changes in pH values of $Fe^0/FeS_2/H_2O$ systems are limited to giving the initial and (more rarely) the final pH values [18,47,53]. However, because Fe-sulfides are expected to induce a pH shift, and $Fe^0$ corrosion is a proton consuming process, only careful real-time in-situ monitoring of the pH value during the course of the experiment can improve the understanding of the system. This corresponds to the approach of Noubactep et al. [55,56,82,88], who used different dosage of $FeS_2$ and showed different final pH values in long-term quiescent batch experiments (up to 120 days). The same authors particularly demonstrated that uranium removal is only quantitative in $Fe^0/FeS_2$ systems having final pH values larger than 4.5. There was no uranium removal in the pure $FeS_2$ system, while the pure $Fe^0$ system also exhibited quantitative uranium removal.

Another important aspect was reported by Mackenzie et al. [45], who were able to decrease the pH of their experimental system by admixing $Fe^0$ with troilite (FeS). The same authors also compared several systems for scavenging dissolved $O_2$, and identified an $Fe^0$/sand mixture as the best mixture. Based on these results, Kenneke and McCutcheon [29] positively tested the $Fe^0/FeS_2$/sand as a further improvement in a pretreatment zone (PTZ). The mix composition of the PTZ (in weight %) was: $Fe^0$ (10), $FeS_2$ (10) and sand (80). These insightful results from both batch and column studies demonstrate the suitability of Fe-sulfide minerals to sustain the efficiency of $Fe^0/H_2O$ systems for water treatment. Sections 5 and 6 present a systematic path to achieve reliable results within the coming few years.

## 5. Extending the Application of $Fe^0/FeS_2$ Systems

Monitoring the behaviors of selected contaminants, including tracers, has been used as a conventional tool to investigate the efficiencies of $Fe^0/H_2O$ systems for water treatment [15,89,90]. The rational selection of model contaminants has been questioned in view of the myriad of species that are potential contaminants worldwide [57,91–94]. The complexity of the $Fe^0/FeS_2/H_2O$ system (Section 4) suggests that its investigation using the methylene blue discoloration method (MB method) [33] would be a very helpful approach with which to accelerate the understanding of the system.

The MB method is summarized by Btatkeu-K et al. [92]. It exploits the low affinity of cationic MB for the positively charged surface of iron corrosion products (FeCPs) under natural conditions (pH > 5.0). Comparatively investigating the efficiency of $Fe^0$/sand mixtures with the pure sand system as a reference enables the relative quantification of the extent of iron corrosion. In fact, the pure sand system is an excellent adsorbent for MB, while sand coated with FeCPs has no affinity for MB. Performing parallel experiments with the same sand and various $Fe^0$ specimens is thus a powerful

tool with which to characterize the $Fe^0$ reactivity. The most reactive specimen produces the largest amount of FeCPs to coat sand under experimental conditions and exhibits the lowest extent of MB discoloration [95]. This simple tool has been adapted to investigate several aspects of the process of contaminant removal in $Fe^0/H_2O$ systems [96–99], and is expected to accelerate the investigation of the $Fe^0/FeS_2$ system. In particular, Btatkeu-K et al. [96] used the MB method to clarify the controversy in the literature concerning the $Fe^0/MnO_2$ system.

Extending the application of $Fe^0/FeS_2$ to drinking water treatment requires a profound understanding of its operating mode. Cui [57] has recently used the MB method to investigate the operating mode of $FeS_2$ in enhancing the efficiency of $Fe^0/H_2O$ systems. The same author used methyl orange (MO) to support the interpretation of the results of MB discoloration. Contrary to MB, MO is an anionic dye exhibiting an excellent affinity to FeCPs. MB and MO are additionally similar in their molecular size. Cui's results [57] clearly demonstrated that $FeS_2$ dissolution lowered the pH value, and neither MB nor MO were quantitatively discolored before the subsequent pH increase reached values larger than 4.5. Additionally, at pH < 4.5, there was no significant difference between MB and MO discoloration. Most importantly, by keeping the pH value low, Cui [57] has irrefutably demonstrated that adsorption and co-precipitation are the fundamental mechanism of contaminant removal in $Fe^0/H_2O$ systems. The pure sand system could not discolor MO, but all other investigated systems discolored both MB and MO.

The results of Cui [57] frontally contradict those of Chen et al. [47], who used the same pyrite, but investigated the discolorations of three different dyes (Orange II, Reactive Red X-3B and Amido Black 10B). Chen et al. [47] is entitled "Pyrite enhanced the reactivity of zero-valent iron for reductive removal of dyes." According to Miyajima and Noubactep [33] the term "reactivity" is misused, and according to the results of Cui [57], the expression "reductive removal" is inappropriate. On the other hand, Chen et al. [47] have not rooted the interpretation of their results on the recorded final pH values. Another weakness of Chen et al. [47] is that all three dyes are anionic in nature, and therefore interact very strongly with in-situ generated FeCPs. In other words, whether the dyes are reductively transformed or not, they are discolored by adsorption and co-precipitation. This last aspect makes the term "discoloration" better than "removal" in investigating dye interactions in $Fe^0/H_2O$ systems [98,99].

There are several ways to extend the efficiency of $Fe^0/H_2O$ system by amending it with pyrite. For example, small amounts of $FeS_2$ can be added to a $Fe^0/sand$ system to sustain iron corrosion in the initial phase of the system operation. The intrinsic reactivity of the selected $FeS_2$ mineral should be known, as well as the kinetics of its long-term dissolution. In other words, there is no need to verify whether pyrite can enhance the efficiency of a $Fe^0/H_2O$ system or not [44,100,101]. Instead, there is a need to rationally select both $Fe^0$ and $FeS_2$ and their relative mixing ratios to achieve a given remediation goal [57]. Similarly, the in-depth mechanisms mediating contaminant removal in $Fe^0/FeS_2$ mixtures are known; thus, the open question is 'how to sustain them in the long-term'?. Another point to note is that, the in-situ dissolution of a $FeS_2$ mineral cannot prevent $Fe^0$ surface passivation because FeCPs are not soluble under environmental conditions. However, $FeS_2$ dissolution locally increases the pH value and delays the precipitation of FeCPs at the surface or in the vicinity of $Fe^0$.

## 6. Knowledge Gaps and Future Directions

A systematic approach is required in the coming years to better understand the operating mode of $Fe^0/FeS_2/H_2O$ systems. Figure 2 presents a conceptual summary of the focal areas of future research critical for the development of $Fe^0/FeS_2/H_2O$-based water treatment systems. Figure 2 also shows the key inputs/outputs of each step, including feedbacks between the various steps, indicating the iterative nature of the design process. Specifically, future research on $Fe^0/FeS_2/H_2O$ systems should include: (i) characterization of the long-term $Fe^0$ and $FeS_2$ reactivities, (ii) determination of the design and operation principles of $Fe^0/FeS_2/H_2O$ systems; (iii) laboratory developments and evaluations of functional prototypes of $Fe^0/FeS_2/H_2O$ systems, including multi-soil layer systems; (iv) pilot-scale and

field testing and evaluation of actual $Fe^0/FeS_2/H_2O$ systems; and (v) outreach and dissemination of the $Fe^0/FeS_2/H_2O$ systems, including monitoring and evaluation by potential end-users of the technology.

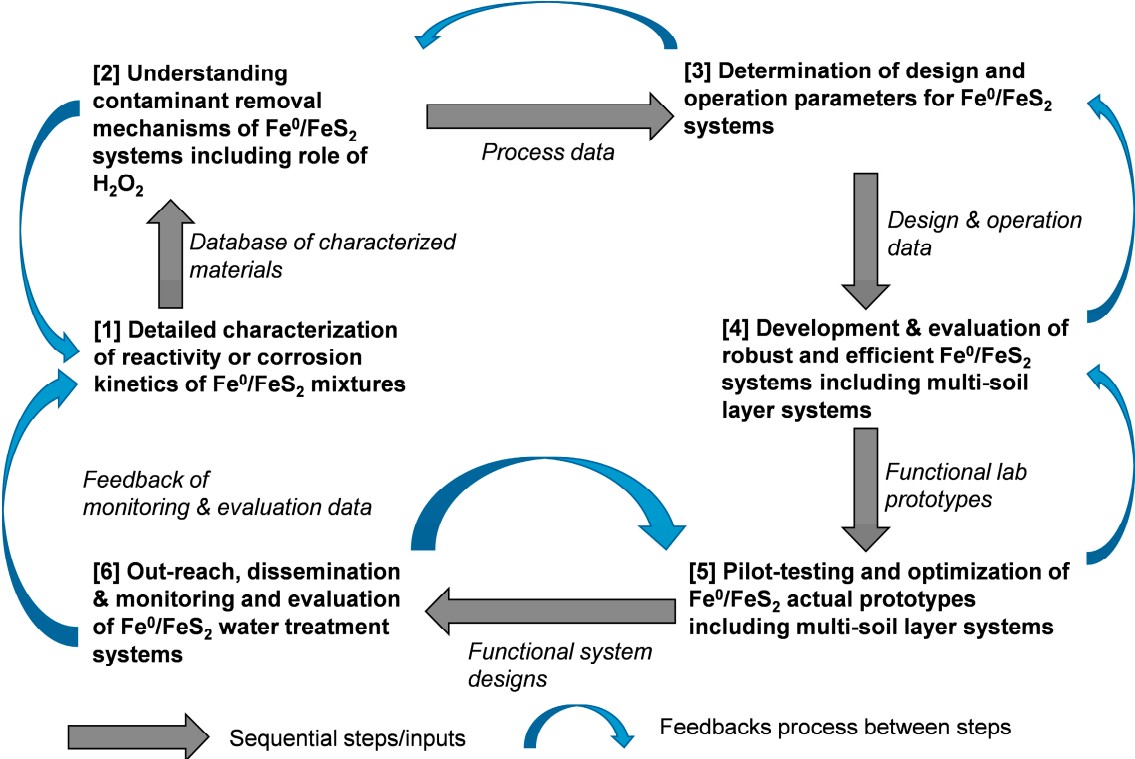

**Figure 2.** A conceptual depiction of proposed roadmap in the development of $Fe^0/FeS_2/H_2O$ water treatment systems, highlighting key thematic areas for future research.

## 6.1. Characterization of Reactivity of Materials

The long-term reactivity of both $FeS_2$ and $Fe^0$ used in $Fe^0/FeS_2/H_2O$ systems are key factors controlling the contaminant removal efficiency of such systems. Yet data on corrosion kinetics is often lacking in most literature focusing on $Fe^0/FeS_2/H_2O$ systems [47,53,102]. On the one hand, highly reactive $Fe^0$ filter media may be effective in contaminant removal in the short-term, but are likely to lose such capacity in the long-term [102]. On the other hand, $Fe^0$ filter materials with low reactivity may have low contaminant removal capacity in the short term, but higher operational longevity. In some water treatment applications, an optimum mixture of $Fe^0$ and $FeS_2$ may need to be determined in order to design effective $Fe^0/FeS_2/H_2O$ systems. In this regard, characterization of the corrosion kinetics of $Fe^0$ and $FeS_2$, and the factors controlling such processes are critical to better understand the contaminant removal capacity of $Fe^0/FeS_2/H_2O$ systems. Such studies should culminate into the establishment of a database of well-characterized $Fe^0$, $FeS_2$ and their optimal mixing ratios.

## 6.2. Design and Operation Principles of $Fe^0/FeS_2/H_2O$ Systems

In reality, water treatment systems are designed to treat raw waters with diverse physico-chemical and biological properties, including salinity, alkalinity, natural organic matter and other potentially-interfering ionic species. Therefore, before $Fe^0/FeS_2/H_2O$ can be widely adopted for drinking water treatment, design engineers and managers for water utilities will require information on the design and operation principles of the $Fe^0/FeS_2/H_2O$ systems for various types of raw waters likely to be encountered. Unlike other competing technologies, such as membrane filtration and reverse osmosis, the design and operation principles for $Fe^0/FeS_2/H_2O$ systems are still lacking. Therefore, further research using both batch and column experiments is required to develop the design and operation

parameters of $Fe^0/FeS_2/H_2O$-based water treatment systems. Subsequently, the design and operation parameters can be used to develop and evaluate prototypes of $Fe^0/FeS_2/H_2O$-based drinking water treatment systems, including those entailing multi-soil layer systems. Such research forms a critical step in the development, and subsequent application of $Fe^0/FeS_2/H_2O$ systems for drinking water treatment.

*6.3. Laboratory-Scale Development Functional Prototypes*

The design principles and data from the previous steps will be used to design laboratory-scale functional prototypes of $Fe^0/FeS_2/H_2O$ systems. The evaluation will entail determination of the effects of operational conditions (e.g., hydraulic loading and residence times) on the capacity of the prototypes to remove target contaminants in synthetic aqueous systems and natural waters. This phase will also include the application of simulation models to better understand system performance. In this regard, the MB method (Section 5) will be very helpful at this stage.

*6.4. Pilot-Scale and Field Testing and Evaluation*

In this step, pilot scale $Fe^0/FeS_2/H_2O$ systems will be designed and evaluated under field conditions to address the scale issues associated with laboratory-scale prototypes. The recent three-layer-design (one $Fe^0$/sand unit sandwiched between two biosand filters) of a household water filter presented by Tepong-Tsindé et al. [27] can be used as starting prototype. In this design, the $Fe^0$/sand unit can be amended with various $Fe^0$ to $FeS_2$ ratios. Evaluation data from this step will be used to optimize and finalize the system designs for the removal of target contaminants, and adequate capacity to treat water/wastewater.

*6.5. Outreach and Dissemination of the $Fe^0/FeS_2/H_2O$ Technology*

The final phase will entail outreach and dissemination of the $Fe^0/FeS_2/H_2O$ technology to the target end-users. The monitoring and evaluation (M&E) system will include system performance evaluations by researchers and utility managers, including perceptions of end-users on the resulting water quality. M&E data from this phase can then be used to further improve the technology, and design improved systems.

## 7. Conclusions

Fe-sulfides minerals could be important for the design of future sustainable $Fe^0$-based water treatment systems, as currently acknowledged. They have already been useful in the following applications: (i) clarifying the mechanism of contaminant removal in $Fe^0/H_2O$ systems, and (ii) scavenging $O_2$ in pre-treatment zones to delay permeability loss and/or sustain the $Fe^0$ reactivity. However, given the many uncertainties coupled with the long-term reactivity of $Fe^0$ (and Fe-sulfides) under environmental conditions, only well-conceived, long-lasting, systematic laboratory and pilot studies would enable the exploitation of the huge potential of $Fe^0$/Fe-sulfides/$H_2O$ for environmental remediation and drinking water supply. A summary of key knowledge gaps and future directions was presented, including the need to adapt $Fe^0$/Fe-sulfides-layers to the design of next generation multi-soil-layering (MSL) systems for decentralized wastewater treatment.

**Author Contributions:** M.X., R.H., X.C., W.G. and C.N. contributed equally to manuscript compilation and revisions. All authors have read and agreed to the published version of the manuscript.

**Funding:** This work is supported by the Ministry of Science and Technology of China through the program "Research on Mechanism of Groundwater Exploitation and Seawater Intrusion in Coastal Areas" (project code 20165037412) and by the Ministry of Education of China through "the Fundamental Research Funds for the Central Universities" (project code: 2015B29314). It is also supported by Jiangsu Provincial Department of Education (project code: 2016B1203503) and Postgraduate Research and Practice Innovation Program of Jiangsu Province (project code: SJKY19_0519, 2019B60214).

**Acknowledgments:** The manuscript was improved by insightful comments of anonymous reviewers from Processes. We acknowledge support by the German Research Foundation and the Open Access Publication Funds of the Göttingen University.

**Conflicts of Interest:** The authors declare no conflict of interest.

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
