# Peer review of "Understanding the Operating Mode of Fe0/Fe-Sulfide/H2O Systems for Water Treatment"

_processes, doi:10.3390/pr8040409_

Round 1
Reviewer 1 Report
This paper provides an interesting insights and suggestions on how Fe/FeS2/H2O2 system works in drinking water treatment. Also, authors have provided: literature overview of this kind of systems, further areas of research (gaps in the field) and steps needed to further develop this kind of systems. After these minor comments are taken into consideration I can suggest this paper to be accepted for publication in journal Processes:
Maybe authors can change title, it is not clear "on what" are they implying. Maybe: "Future of Fe0/Fe-sulfide/H2O systems for water treatment" or something similar.
Ref. 62 is not correct, authors from ref. 62 did not use Ca2+ to remove F-. They used Fe0 to remove it.
Table 2. Google scholar is probably not the beast tool to measure current bibliometry or any kind of real scientific impact. Web of Science Core Collection or SCI are more appropriate tools for this.
Author Response
This paper provides an interesting insights and suggestions on how Fe/FeS2/H2O system works in drinking water treatment. Also, authors have provided: literature overview of this kind of systems, further areas of research (gaps in the field) and steps needed to further develop this kind of systems. After these minor comments are taken into consideration I can suggest this paper to be accepted for publication in journal Processes:
Many thanks for this evaluation!
Maybe authors can change title, it is not clear "on what" are they implying. Maybe: "Future of Fe0/Fe-sulfide/H2O systems for water treatment" or something similar.
Many thanks, we have changed to “Understanding the operating mode of Fe0/Fe-sulfide/H2O systems for water treatment”
Ref. 62 is not correct, authors from ref. 62 did not use Ca2+ to remove F-. They used Fe0 to remove it.
Many thanks for this remarks. The authors or ref. [62] have made this demonstration in rationalizing why Fe0 could be successful were precipitation with Ca2+ has failed.
Table 2. Google scholar is probably not the best tool to measure current bibliometry or any kind of real scientific impact. Web of Science Core Collection or SCI are more appropriate tools for this.
Many thanks, we have now used SCOPUS.
Reviewer 2 Report
Applying metallic iron (Fe) is important for water treatment systems and us.
This manuscript is summarized a lot of interesting views.
Please make sure my comment.
1) P.4, Figure 1
It is difficult to understand the letters in the figure 1.
Could you please write the letters more clearly in this figure.
Author Response
Applying metallic iron (Fe) is important for water treatment systems and us. This manuscript is summarized a lot of interesting views.
Many thanks for this evaluation!
1) P.4, Figure 1
It is difficult to understand the letters in the figure 1.
Could you please write the letters more clearly in this figure.
Many thanks, we have redrawn Figure 1.